# Unleashed Treasures of Solanaceae: Mechanistic Insights into Phytochemicals with Therapeutic Potential for Combatting Human Diseases

**DOI:** 10.3390/plants13050724

**Published:** 2024-03-04

**Authors:** Saima Jan, Sana Iram, Ommer Bashir, Sheezma Nazir Shah, Mohammad Azhar Kamal, Safikur Rahman, Jihoe Kim, Arif Tasleem Jan

**Affiliations:** 1School of Biosciences and Biotechnology, Baba Ghulam Shah Badshah University, Rajouri 185234, Jammu and Kashmir, India; saimajan.scholar@bgsbu.ac.in (S.J.); shahsheezma@gmail.com (S.N.S.); 2Department of Medical Biotechnology, Yeungnam University, Gyeongsan 712-749, Republic of Korea; sanairam157@yu.ac.kr; 3Department of School Education, Srinagar 190001, Jammu and Kashmir, India; ommerbashir@gmail.com; 4Department of Pharmaceutics, College of Pharmacy, Prince Sattam Bin AbdulAziz University, Alkharj 11942, Saudi Arabia; ma.kamal@psau.edu.sa; 5Department of Botany, Munshi Singh College, BR Ambedkar Bihar University, Muzaffarpur 845401, Bihar, India; shafique2@gmail.com

**Keywords:** Solanaceae, bioactive compounds, pharmaceuticals, phytochemistry, therapeutics

## Abstract

Plants that possess a diverse range of bioactive compounds are essential for maintaining human health and survival. The diversity of bioactive compounds with distinct therapeutic potential contributes to their role in health systems, in addition to their function as a source of nutrients. Studies on the genetic makeup and composition of bioactive compounds have revealed them to be rich in steroidal alkaloids, saponins, terpenes, flavonoids, and phenolics. The Solanaceae family, having a rich abundance of bioactive compounds with varying degrees of pharmacological activities, holds significant promise in the management of different diseases. Investigation into Solanum species has revealed them to exhibit a wide range of pharmacological properties, including antioxidant, hepatoprotective, cardioprotective, nephroprotective, anti-inflammatory, and anti-ulcerogenic effects. Phytochemical analysis of isolated compounds such as diosgenin, solamargine, solanine, apigenin, and lupeol has shown them to be cytotoxic in different cancer cell lines, including liver cancer (HepG2, Hep3B, SMMC-772), lung cancer (A549, H441, H520), human breast cancer (HBL-100), and prostate cancer (PC3). Since analysis of their phytochemical constituents has shown them to have a notable effect on several signaling pathways, a great deal of attention has been paid to identifying the biological targets and cellular mechanisms involved therein. Considering the promising aspects of bioactive constituents of different Solanum members, the main emphasis was on finding and reporting notable cultivars, their phytochemical contents, and their pharmacological properties. This review offers mechanistic insights into the bioactive ingredients intended to treat different ailments with the least harmful effects for potential applications in the advancement of medical research.

## 1. Introduction

Plants have long been known as an excellent source of bioactive compounds with unique pharmacological properties [1,2,3]. Being a good source of phytochemicals such as alkaloids, terpenoids, and phenylpropanoids, they can be used to treat different illnesses, in addition to providing a source for the development of new medicines [4,5,6]. *Solanum* L. (Solanaceae; also referred to as nightshades) is one of the largest genera of the Solanaceae family, distributed across tropical, subtropical, and temperate regions [7,8]. The members of the Solanaceae family, being annual, biennial, or perennial, are herbaceous and exhibit great floristic diversity, phytochemical characteristics, and ethnobotanical significance. Solanum family members have yielded a variety of pharmacologically active compounds with distinct roles such as antirheumatic, antimicrobial, antioxidant, and anti-tumor [9]. It is an important plant taxon with species known for their therapeutic value [10]. Of the 670 compounds that have been reported to be found in the Solanum genus, including steroidal saponins, alkaloids, pregnane glycosides, terpenes, flavonoids, lignans, sterols, phenols, and coumarins, steroidal alkaloids such as solasodine, solasonine, and solamargine are of particular interest due to their unique medicinal properties, which have been shown to improve human health [11,12]. The chemical composition of many of these species is still poorly understood or has yet not received much attentionm, but they have been documented to have immense pharmacological potential, which offers greater hope for the development of innovative medicine effective against a wide range of human illnesses. Against this background, the present study provides a thorough evaluation of the morphological characteristics, regional distribution, and secondary metabolites with unique pharmacological capabilities across seven different species of the Solanum genus. Special attention has been paid to elucidating the mechanisms by which they modulate signaling across diverse signaling pathways to control the course of cells toward apoptotic and/or autophagic processes.

## 2. Distribution and Morphology

The Solanaceae family has 97 genera and about 2700 species, distributed across all continents except Antarctica [13]. The greatest diversity has been reported in Central and South America [14,15]. It is commonly distributed in the equatorial zone of Ibero-America. It inhabits distinct habitats (wastelands, old fields, ditches, cultivated land, overgrazed grass fields, railway cuttings, areas near buildings, riverbanks, etc.), from deserts to rainforests, and establishes itself in the lower vegetation and conquers distraught areas [16,17]. Solanaceae is ranked the third most economically significant family after Poaceae and Fabaceae, encompassing a vast array of species and boasting the most abundant collection of edible varieties within its genus Solanum. Several species of Solanum are economically important at a global scale, with several major agricultural crops (*S. tuberosum* L., potato; *S. lycopersicum* L., tomato; *S. melongena* L., eggplant), and the genus also contains locally important fruit crops [18]. Furthermore, the closely related Capsicum genus features widely consumed vegetables such as peppers [19,20,21]. Some of the medicinally important plants of the genus Solanum are *Solanum nigrum*, *Solanum torvum*, *Solanum indicum*, *Solanum surattense*, *Solanum Villosum*, *Solanum viarum*, and *Solanum incanum* (Figure 1).

***S. nigrum***, an annual herb commonly found in Europe, Asia, and Africa, exhibits a widespread distribution worldwide, spanning from sea level to elevations exceeding 11,482 feet [22,23]. It particularly thrives in moist environments and demonstrates strong adaptability to soils rich in nitrogen or phosphorus [24]. *S. nigrum* typically reaches a height of 30–120 cm. The leaves are ovoid in shape with a trilateral base, measuring 4–7 cm in length and 2–5 cm in width. The upper portion of the leaves has a smooth texture. The flowers of *S. nigrum* are whitish in color, with yellow anthers [25]. The stem of the plant is usually bifurcated (branching into two) and covered with fine hairs. Its inflorescence is arranged into extra-axillary umbels and the sepals have a cup-like shape. The elliptical anthers measure approximately 2.5–3.5 mm, while the filaments are 1.5 mm in length. At maturity, the fruits of *S. nigrum* turn dull black in color and are spherical in shape, with a diameter of 8–10 mm [26].

***Solanum torvum*** is as an erect shrub, reaching a height of 1–3 m and characterized by a prickly stem [27]. The young stem and branches exhibit a vibrant green color and are adorned with delicate trichomes, while the bark of its mature stems takes on a brown to dark gray color. The leaves of *S. torvum* maintain their green color throughout the year and possess a broadly ovate structure, measuring between 5 and 21 cm in length and 4 and 13 cm in width. The leaf margin is usually intact but may occasionally have up to seven broad triangular lobes. Both the upper and lower leaf surfaces are covered with fine stellate hairs and there are scattered prickles along the main veins. The upper surface of the leaf is darker in color compared to the lower surface. The base of the leaf lamina is truncated and oblique, while the apex is acute to acuminate. The leaves of *S. torvum* are characterized by densely stellate-pubescent petioles, measuring 1–5.5 cm in length, with some curved prickles reaching up to 10 mm in length. The inflorescences (consist of 50–100 flowers) are dense and compact and covered with trichomes that grow up to 6 cm long and branching 1–4 times. The flowers are pentameric, with slender and hairy sepals that are 2–3 mm long, forming a calyx that measures 4–6 mm in length. The petals are white to cream in color, approximately 1 cm long, and together they form a stellate corolla with a diameter of 1.5–2 cm. The androecium is composed of connivant stamens that are yellow in color, with anthers measuring 6–8 mm in length and about 1 mm in width. The gynoecium consists of a conical ovary that is glandular, topped with a 10–12 mm long glabrous style, which ends with a capitate stigma protruding over the androecium. The fruits of *S. torvum* are globular berries, measuring 1–1.5 cm in diameter. When ripe, they appear pale grayish–green and contain a few to several flat woody seeds, which are 1.5–2 mm long.

***Solanum indicum*** is known to occur in regions such as India, Sri Lanka, Malaya, China, and the Philippines, spanning from sea level up to approximately 1500 m above sea level [16]. The plant is a biennial herb or small shrub with upright growth and spiky characteristics. Its sturdy stems feature large, sharp prickles that have a long compressed base, often with a slightly curved shape. The plant’s leaves are oblong in shape, measuring 5–15 cm in length and 2.5–7.5 cm in width. They have a sub-entire margin with a few triangular–oval lobes and sparse prickles and hair on both sides. The leaf base is cordate or truncate, occasionally unevenly shaped, and the petioles are hairy on both sides and range from 1.3 to 2.5 cm in length. The flowers are grouped into racemose extra-axillary cymes, which are held up by short peduncles covered in stellate hairs. The corolla, with a length of about 0.8 cm, displays a pale purple hue and is adorned with darker purple stellate hairs on its outer surface. The lobes of the corolla measure roughly 5 mm. The stamens, connected to the corolla, feature brief filaments and large anthers, constituting the male reproductive components of the flower. The fruits, or berries, are globose with a diameter of ~70.8 cm when fully ripe. They are glabrous and have a dark yellow color at maturity. The seeds are about 0.4 cm in diameter and exhibit minute pits.

***Solanum surattense*** is a perennial herbaceous plant that is widely distributed in Australia, Ceylon, India, Malaysia, Polynesia, and Southeast Asia [28]. The species typically reaches a height of about 1.2 m and has a woody base. The stem is highly branched, exhibiting a zigzag pattern, with young branches covered in dense satellite and tomentose hairs. The straight, glabrous, and shiny prickles on the stem are often 1–3 cm in length. The leaves are ovate–elliptic, deeply lobed, and tapered at the base, with spines present along the veins and margins. The flowers are bluish pink and arranged into extra-axillary racemes. The calyx has five free lobes that are ovate and prickly. The corolla is broadly ovate–triangular, with five acute lobes. The plant produces globose berries that are initially green with white stripes but turn yellow when ripe. The seeds are circular and numerous and have a smooth surface [29].

***Solanum villosum*** is found in the Euro-Siberian, Irano-Turanian, and Mediterranean regions of the globe [30]. It shows distribution in countries such as Africa, Kenya, China, India, and Pakistan [31,32]. *S. villosum* possess a brittle stem that can reach a height of ~1 m [33]. The leaves of *S. villosum* are rhombic to ovate–lanceolate, measuring 2.0–7.0 cm in length and 1.5–4.0 cm in width. The leaf margins can be either smooth or slightly wavy with shallow teeth. The inflorescence is simple, forming umbellate or loosely arranged solitary cymes. The calyces are approximately 1.2–2.2 mm long and slightly curved and may bend downward or adhere to the base of mature berries. The berries themselves are usually longer than they are wide and orange in color and have a width of 6–10 mm. When fully ripe, the berries detach from the calyces and fall off [34].

***Solanum viarum*** has its origin in Brazil and Argentina but exhibits widespread distribution as a weed throughout South America, Africa, India, Nepal, the West Indies, Honduras, Mexico, Cameroon, and the Democratic Republic of the Congo [35,36]. Both agricultural and natural environments have been occupied by *S. viarum* [37,38]. The plant’s spiky prickles make handling them challenging. The prickles, which are dispersed on the stem and the leaves, are about 1–2 cm long and range in color from white to yellowish. The leaves are alternate, have a broad end at the base, and, especially on young leaves, have somewhat wavy borders. *S. viarum* leaves are pubescent, 10–20 cm long, 6–15 cm wide, and deeply split into broad pointed lobes. The upper side of the leaf is gray–green and the lower surface is greenish white [38]. The flowers have cream-colored filaments and are white in color [39]. The unripe fruits exhibit white mottling patterns and undergo a color transformation to yellow upon ripening. They are smooth and round, measuring approximately 2–3 cm in diameter. Each fruit contains a range of 190–385 light red–brown seeds, with a diameter of 2.2–2.8 mm [36,40].

***Solanum incanum*** is indigenous to the Horn of Africa and has a broad distribution in that region. It is characterized by its thorny leaves, yellow fruits, and blue flowers with yellow pistils [11,41]. The plant reproduces through seed propagation and the germination process is relatively slow. Table 1 summarizes the basic features of the above-mentioned species of the genus Solanum.

## 3. Secondary Metabolites from Solanaceae

Primary and secondary metabolites are the two categories of organic compounds/metabolites produced by plants [47,48]. Secondary metabolites (SMs) are small organic compounds with molecular masses less than 3000 da that develop from the primary metabolites during the embolism of plants. Different plant species have different metabolite types and compositions. In a recent study, Satish et al. [49] pointed out that almost 200,000 SMs have been identified and described. SMs are fascinating due to their structural diversity and their potency as therapeutic candidates and/or antioxidants, among other diverse reasons [50]. Almost 30% of pharmaceutical drugs arise directly or indirectly from secondary metabolites [51]. Based on the biosynthetic route, we find three main categories of plant metabolites: phenolic groups, composed mainly of simple sugars and benzene rings; terpenes and steroids, primarily consisting of carbon and hydrogen; as well as alkaloids, which incorporate nitrogen [52]. The SMs produced by different members of the Solanaceae family include diosgenin, solamargine, solanine, apigenin, etc. The SMs reported in Solanaceae exhibit a wide range of biological roles, including antioxidant, anticarcinogenic, and anti-inflammatory activities [53]. Broadly, they are known to influence several crucial cellular processes that affect cell cycle progression, alter membrane permeability, and induce apoptosis. SMs with antiproliferative properties have been found to influence several crucial cellular processes: for example, they can affect the progression of the cell cycle, leading to cell cycle arrest and preventing uncontrolled cell division. These compounds can also modulate the expression of certain genes involved in cancer cell survival and growth. The SMs present in the Solanaceae family have demonstrated the capacity to trigger apoptosis (programmed cell death) across diverse forms of cancer cells. These compounds activate different signaling pathways, which can lead to cell death in cancer cells. The differences in their effects on cancer cells may arise from variations in the chemical structure of the compounds and the specific sensitivity of different types of cancer cells. They can interfere with various signaling pathways that regulate cell survival and proliferation and can induce changes in the mitochondrial membrane, which may lead to mitochondrial dysfunction and apoptosis. They can alter the metabolic processes in cancer cells, affecting their energy production and survival. The diversity of effects observed in different cancer cell types can be attributed to the complex interactions between these SMs and the specific molecular characteristics of the cancer cells.

### 3.1. Diosgenin

Diosgenin, identified as 25R-spirost-5-en-3β-ol, is a derivative resulting from the hydrolysis of dioscin present in the rootstock of yam (Dioscorea). It is prevalent in natural plant sources in the glucoside form [54]. Diosgenin, a steroidal sapogenin, is present in a variety of plants, including Solanum species, namely Dioscorea nipponoca, Costus speciosus, and Trigonella foenum graecum [55,56]. This biologically active phytochemical plays a significant role in various plant actions, including its involvement in functional food properties [54]. It serves as a common starting point in the synthesis of steroids and contraceptives. Additionally, it finds application in medicinal forms for treating conditions such as leukemia, hypercholesterolemia, climacteric syndrome, and colon cancer [56]. Furthermore, diosgenin is employed as the primary material for the synthesis of oral contraceptives, sex hormones, and various other steroidal compounds [54]. Within plants, it exists in the configuration of steroidal saponins. The anti-tumor effects of diosgenin are attributed to its involvement in multiple mechanisms (inhibiting tumor cell growth, metastasis, and invasion, promoting the apoptosis of tumor cells, and blocking the cell cycle), with actions on different targets and pathways (Figure 2).

#### 3.1.1. Inhibiting tumor cell growth, metastasis, and invasion

Diosgenin reduces the expression of matrix metalloproteinases (MMPs), including MMP-2 and -9, which are critical for tumor cell migration and invasion [57]. By inhibiting MMPs, diosgenin interferes with the metastatic process of tumor cells. Diosgenin exhibits significant effects in inhibiting tumor growth and metastasis via a reduction in the phosphorylation of IKKβ and NF-κB, leading to the inhibition of TNF-α and IL-6 production in endothelial cells. Furthermore, it inhibits the synthesis of endothelin-1 (ET-1) and plasminogen activator inhibitor 1 (PAI-1) within the endothelial cells, subsequently reinstating insulin-mediated vasodilation [58]. 

Angiogenesis plays a vital role in the progression of tumors, facilitating their growth, metastatic spread, and invasion into surrounding tissues [59]. Vascular endothelial growth factor (VEGF) serves as a pivotal controller of the angiogenic process and differentiation, playing a vital role in various angiogenic processes [60]. VEGF facilitates tumor growth by providing nutrition and waste excretion pathways for tumors and creating routes for tumor cells to enter the circulatory system through binding to the corresponding receptor, VEGFR. Moreover, the vascular endothelium plays a significant role in orchestrating both angiogenesis and vascular tone, constituting a specialized monolayer of cells with distinct site-specific functions and morphology [61]. Diosgenin has demonstrated the capability to impede angiogenesis by targeting several signaling pathways, encompassing HIF-1α, GRP78, VEGFR, PI3K/AKT, ERK1/2, and FAK. Additionally, it diminishes the expression of the VEGF and fibroblast growth factor 2 (FGF2) proteins, restraining the tubular formation of endothelial cells within cancer cells, consequently exerting a suppressive effect on angiogenesis [62]. These findings highlight the potential of diosgenin as a therapeutic agent to impede tumor angiogenesis, which could have significant implications in cancer treatment and management. By targeting VEGF and the vascular endothelium, diosgenin can effectively inhibit tumor angiogenesis, which is crucial for tumor growth and metastasis.

#### 3.1.2. Regulation of the Apoptosis Pathway 

Apoptosis is a well-featured process of programmed cell death, characterized by cell shrinkage and fragmentation [63]. In the apoptotic process, chromosomal DNA undergoes fragmentation (a process mediated by specific endonucleases) and the DNA is cleaved into oligonucleosomal-size fragments [64]. Apoptosis can be triggered through two principal pathways: the extrinsic (transmembrane) pathway and the intrinsic (mitochondrial) pathway. Recently, a novel lysosomal pathway of apoptosis has been identified [65,66]. Several factors, such as lysosomotropic photosensitizer agents, oxidative stress, oxidized lipids, serum starvation, DNA damage, and the activation of Fas and TNF-α, initiate the lysosomal pathway. This pathway encompasses the liberation of lysosomal cathepsins. Lysosomal membrane permeabilization, a part of this process, can result in both caspase-dependent and caspase-independent forms of apoptosis-like cell death, and in certain instances, it may lead to necrosis.

Diosgenin prompts apoptosis by elevating the presence of cytochrome C within the cytosol, along with increased levels of cleaved-caspase-3 and cleaved-PARP1. This is accompanied by an increased Bax/Bcl-2 ratio [67]. Within the HepG2 cells, diosgenin initiates apoptosis by engaging the mitochondria/caspase-3-dependent pathway, mediated by the Bcl-2 protein family. This family is implicated in tumor progression, with changes in the expression of its constituents serving as prognostic indicators across diverse lymphoid malignancies [68]. Caspases, categorized as cysteinyl aspartate-specific proteinases and belonging to the interleukin-1β-converting enzyme family, hold pivotal roles as regulators of apoptosis in eukaryotic cells. They are important for governing processes such as cell growth, differentiation, and apoptosis [69]. Their activation and function within the caspase cascade system are regulated by various molecules, including the inhibitor of apoptosis proteins, Bcl-2 family proteins, calpain, and Ca^2+^. The p53 gene, a well-known tumor suppressor gene, is frequently subject to mutations and inactivation in the development of most human cancers [70,71]. Diosgenin triggers apoptosis in different cancer cell types, including bladder cancer, K562, rectal cancer, and glioblastoma cancer cells, through the induction of DNA damage. This outcome is linked to substantial increases in the protein levels of Bak, Bax, Bid, p53, caspase-3, and caspase-9. Simultaneously, there is a reduction in the protein levels of Bcl-2, Bcl-xL, survivin, and RNA [72,73]. One of the key targets of endoplasmic reticulum stress is calcium homeostasis, and the disruption of Ca^2+^ homeostasis and mitochondrial dysfunction play crucial roles in apoptosis [74]. Diosgenin triggers erythrocyte contraction and the disturbance of erythrocyte membrane phospholipids, leading to Ca^2+^ influx, oxidative stress, and ceramide formation [75]. By regulating calcium homeostasis, diosgenin can induce mitochondrial-dependent cell apoptosis, causing Ca^2+^ release, reducing the intracellular Ca^2+^ concentration, and inducing cell cycle arrest and apoptosis [76].

#### 3.1.3. Blocking the Cell Cycle

Diosgenin’s anti-tumor mechanism involves blocking the tumor cell cycle. It causes cell cycle arrest at various phases, including G2/M, S, and G0/G1, by regulating cyclin B1, p21cip1/Waf1, cdc2, pRb, E2F, CDK2/4/6, CyclinD1, p21, and p27 [73,77]. These regulatory effects lead to the suppression of tumor cell proliferation across various cancer types, including osteosarcoma, hepatoma, chronic myeloid leukemia, and pancreatic cancer.

#### 3.1.4. Regulating Gene Expression and Proteins

Diosgenin has been found to upregulate the autophagy marker LC3 protein and induce autophagy in the hepatoma cells by reducing the ratio of LC3-I to LC3-II [78]. Additionally, diosgenin regulates various molecular pathways in different cancer types. In gastric, lung, and breast cancers, diosgenin modulates the expression of p21 mRNA, nuclear factor UUB activation, prostaglandin E2 synthesis, the miR-145 expression level, and the methylation status. Furthermore, it reduces the expression of cyclooxygenase (COX)-2 and microsomal prostaglandin E synthase (MPGES)-1, while inducing a substantial increase in miR-34a expression. Additionally, it downregulates the genes targeted by miR-34a, including E2F1, E2F3, and CCND1, resulting in the suppression of cancer cell proliferation [4,79]. Diosgenin inhibits the telomerase activity in lung cancer cells (A549) by downregulating the expression of the telomerase reverse transcriptase gene HTERT. Within glioblastoma cells, diosgenin prompts cell differentiation while decreasing cell dedifferentiation. This is achieved through an elevation in the expression of glial fibrillary acidic protein (GFAP), a marker of differentiation, and a concurrent reduction in the levels of dedifferentiation markers such as Id2, N-Myc, TERT, and Notch-1 [80]. Diosgenin demonstrates an anticancer impact by suppressing the expression of mesoderm posterior 1 (MESP1) within gastric cancer cells. This action stimulates processes like cell proliferation, apoptosis, and overall growth inhibition [81].

#### 3.1.5. Regulating Cell Signaling

The signal pathways play a crucial role in mediating the anti-tumor effects of diosgenin. These pathways encompass the mitogen-activated protein kinase (MAPK) signaling pathway, which includes cascades like c-Jun N-terminal kinase (JNK) and extracellular signal-regulated kinase (ERK), as well as p38-MAPK and ERK5. The MAPK/ERK pathway is implicated in diverse cellular functions, including cell proliferation, differentiation, migration, senescence, and apoptosis. Another crucial pathway is the phosphatidylinositol-3 kinase (PI3K)/Akt pathway, which plays a role in cell survival, proliferation, and growth. The nuclear factor kappa B (NF-κB)/STAT3 pathway, which plays a role in regulating inflammation and cell survival. The Wnt/β-catenin signal pathway is responsible for cell fate determination and cell proliferation. The cAMP/PKA/CREB pathway is associated with cell signaling and gene expression regulation. By targeting these diverse signal pathways, diosgenin exhibits its anti-tumor effects by influencing various cellular processes, providing a comprehensive approach to combating tumor development and progression [82]. Diosgenin exhibits its anti-tumor effects through various mechanisms in different types of cancer cells. In colon cancer, esophageal, and osteosarcoma cells, diosgenin governs critical aspects like proliferation, apoptosis, migration, and invasion by obstructing the initiation of the epithelial–mesenchymal transition (EMT). This modulation involves the manipulation of EMT-associated proteins like transforming growth factor β1, E-cadherin, and vimentin. Diosgenin also triggers apoptosis in A549 cells through its impact on proteins within the MAPK signal pathway, including caspase 8, 9, and 3 [83]. Diosgenin downregulates the MAPK, NF-κB, and Akt signaling pathways by inhibiting the phosphorylation of NF-κB/p65, JNK, IKK-β, Akt, and ERK, resulting in its anti-tumor effect. In pancreatic tumor cells, diosgenin effectively inhibits the ERK, JNK, and PI3K/AKT signaling pathways [84]. In human hepatocellular carcinoma cells, diosgenin inhibits NF-κB activity and the activation of NF-κB/STAT3, leading to a significant decrease in the expression of various oncogene products and the inhibition of cell proliferation [85]. In MG-63 cells, diosgenin stimulates both cell proliferation and differentiation by impeding the Wnt/β-catenin signaling pathway, which leads to a decrease in the count of calcified nodules and a restraint in the expression of osteopontin and osteocalcin [86]. Diosgenin inhibits aerobic glycolysis in colorectal cancer cells by regulating glucose transporters (GLUT3 and GLUT4) and inhibiting CREB phosphorylation through the cAMP/PKA/CREB pathway, leading to apoptosis [87]. These diverse pathways and mechanisms collectively contribute to the anti-tumor effects of diosgenin, highlighting its potential as a multi-target and multi-pathway therapeutic agent in cancer treatment.

### 3.2. Solamargine

Solamargine is classified as a glycoalkaloid, which is a type of chemical compound that contains both sugar and an alkaloid. The compound has been isolated and identified from various plants, with structural variations depending on the specific plant source. Solamargine has been investigated for its potential pharmacological properties, including anti-inflammatory, antiviral, and anticancer activities. Some studies suggest that solamargine may exhibit cytotoxic effects on certain cancer cells, making it a subject of interest in cancer research [88,89].

#### 3.2.1. Solamargine-Mediated Cytotoxicity

Solamargine has demonstrated remarkable cytotoxic activity against various cancer cell lines, making it one of the most potent cytotoxic compounds in this plant family [90]. Numerous studies utilizing diverse methodologies consistently demonstrate the cytotoxic effects of solamargine on an extensive array of cancer cell lines. These encompass human liver cancer lines such as HepG2, Hep3B, H22, and SMMC-7721; human lung cancer lines including A549, H441, H520, H661, H69, and H1650; human breast cancer lines like HBL-100, SK-BR-3, ZR-75-1, and MCF-7; human myelogenous leukemia line K562; squamous carcinoma line KB; human prostate cancer line PC3; osteosarcoma lines U2OS, MG-63, and Saos-2; murine melanoma line B16F10; human colon carcinoma line HT29; human cervical adenocarcinoma line HeLa; and human glioblastoma lines MO59J, U343, and U251. Notably, solamargine exhibits minimal toxicity toward normal cell lines, including Chinese hamster lung fibroblasts (V79), human lung fibroblasts (GM07492A), human liver cells (HL-7702), and the human retinal pigment epithelium (RPE1). The cytotoxic effects of solamargine have been found to be dependent on both the dosage and exposure time [91,92,93]. The preceding evidence supports the notion of the heightened cytotoxicity of solamargine specifically against lung cancer cell lines, while demonstrating its minimal impact on normal cells.

#### 3.2.2. Solamargine in the Apoptotic Process

The intrinsic mitochondrial pathway of apoptosis is intricately governed by maintaining equilibrium between the proapoptotic and anti-apoptotic members within the Bcl-2 protein family [94]. The Bcl-2 family proteins play a critical role in regulating the mitochondrial membrane permeability and the intrinsic apoptotic pathway. This family is composed of two main groups: proapoptotic proteins, which promote cell death, and anti-apoptotic proteins, which inhibit apoptosis. Some of the proapoptotic Bcl-2 family proteins include Bcl-10, Bax, Bak, Bik, Bid, Bim, and Bad. These proteins are involved in promoting the mitochondrial membrane’s permeability, which allows the release of proapoptotic factors, such as cytochrome c, into the cytosol. This, in turn, triggers the caspase cascade and leads to apoptosis. On the other hand, the anti-apoptotic Bcl-2 family proteins, such as Bcl-2, Bcl-XS, Bcl-XL, Bcl-x, Bcl-w, and BAG, are important in maintaining the mitochondrial membrane’s integrity and preventing the release of proapoptotic factors [95]. Li et al. [96] demonstrated that solamargine induced apoptosis through mechanisms involving the upregulation of Bax and p53. The increased expression of Bax, a proapoptotic member of the Bcl-2 family, promotes apoptosis by facilitating the permeabilization of the mitochondrial membrane and releasing proapoptotic factors, such as cytochrome c, into the cytosol, where it becomes a key trigger for programmed cell death [97]. Cytochrome c oxidase (Cyt c or Complex IV, EC 1.9.3.1) is an enzyme crucially involved in ATP synthesis and is normally associated with the inner membrane of the mitochondrion, playing a vital role in supporting cellular life.

Specifically, the anti-apoptotic Bcl-xL and Bcl-2 genes were downregulated in solamargine-treated H44, H520, H661, H69, K562, A549, ZR-75-1, SK-BR-3, B16F10, HBL-100, HT29, HepG2, MCF-7, U343, HeLa, MO59J, and U251 cells. The downregulation of these anti-apoptotic genes promotes apoptosis by reducing their inhibitory effect on the process. Furthermore, an increase in the expression of cleaved caspase-3 and caspase-9 was observed in solamargine-treated K562, B16F10, HT29, MCF-7, HeLa, HepG2, MO59J, U343, and U251 cells [98,99]. The presence of cleaved caspase-3 and caspase-9 indicates the activation of the caspase cascade, which is a hallmark of the execution phase of apoptosis. Overall, these findings suggest that solamargine induces intrinsic mitochondrial apoptosis by altering the expression of Bcl-2 family members, leading to the activation of caspases and subsequent programmed cell death in the treated cancer cells.

The extrinsic (transmembrane) pathway of apoptosis is initiated when external death ligands, like the Fas ligand or TNF-related apoptosis-inducing ligand (TRAIL), attach to their respective death receptors located on the cell membrane. This interaction triggers the activation of caspase enzymes, which play a crucial role in coordinating the process of cellular self-destruction. The extrinsic pathway plays a crucial role in regulating cell death and is often implicated in the immune system’s defense against abnormal or infected cells. A recent experiment demonstrated that solamargine induced apoptosis in K562 cells by causing early lysosomal destabilization, followed by subsequent mitochondrial damage. This damage was characterized by an overload of Ca^2+^, a decrease in the membrane potential, and the release of Cyt c. Additionally, it has been observed that solamargine can induce permeabilization of the lysosomal membranes in cancer cells, leading to an increase in the influx of water. Therefore, lysosomes undergo swelling, ultimately resulting in the formation of vacuoles [98]. Enlarged lysosomes contribute to an elevation in surface tension, a reduction in lysosomal integrity, and eventual fragmentation. Furthermore, in the event of cellular damage, repair of enlarged lysosomes becomes more challenging compared to repairing smaller ones [100].

#### 3.2.3. Cell Cycle Arrest

The cell cycle consists of four widely acknowledged stages, i.e., the G1, S, G2, and M phases [101]. Solamargine has demonstrated the capacity to elevate the ratio of the apoptotic sub-G1 peak within several cell lines, including A549 [93,102], H661, H441, H520, and H69 [103], as well as MCF-7, HBL-100, and SK-BR-3 [90], along with Hep3B cells [91,92]. This effect exhibits a dependence on both time and concentration. Conversely, the administration of solamargine led to a reduction in the G2/M phase in HBL-100, MCF-7, SK-BR-3 [90,104], Hep3B [92], SMMC-7721 [105], ZR-75-1 [104], and A549 cells [93,102]. Furthermore, investigations revealed no significant alterations in the G0/G1 phase in ZR-75-1, HBL-100, and SK-BR-3 cells following treatment with solamargine [104]. However, contrary to the previous findings, a more recent study demonstrated that solamargine did not induce cell cycle arrest in K562 cells [106]. The malignant phenotype of cancer cells is primarily attributed to uncontrolled cell cycle regulation, leading to abnormal cell growth and proliferation. Mitogens can impose inhibitory effects on the progression of the cell cycle by inducing the activation of G1-S cyclin-dependent kinase (CDK) activities. This activation subsequently initiates the phosphorylation of the retinoblastoma protein (pRB). The function of pRB is frequently impaired in cancer cells, contributing to their uncontrolled proliferation [107]. Tumorigenic mutations have been identified as the underlying factors behind the malfunctioning of the regulatory mechanisms that control the progression of the cell cycle. These mutations have been observed in diverse tumor types and can disrupt various mitogenic signaling pathways. These pathways encompass the HER2 gene and downstream signaling networks like the PI3K-Akt or Ras-Raf-MAPK pathways, as well as genes that play a role in regulating the cell cycle [108,109]. These alterations disrupt the normal regulation of the cell cycle, leading to uncontrolled cell growth and the proliferation characteristic of cancer cells. The proto-oncogene HER-2/neu plays a crucial role as a regulator of cell proliferation. Studies have shown that solamargine treatment resulted in an upregulation of HER2/neu and topoisomerase II α (Topo-IIα) expression in MCF-7, HBL-100, and SK-BR-3 cells [90].

#### 3.2.4. Regulating Gene Expression and Proteins

Solamargine has been shown to induce apoptosis in cancer cells by modulating the expression of apoptosis-related genes and proteins. Treatment with solamargine has been found to influence the gene and protein expression of specific receptors and signal proteins associated with apoptosis in various cancer cell lines. In the HBL-100, SK-BR-3, ZR-75-1, A549, Hep3B, B16F10, HT29, MCF-7, MO59J, U343, HeLa, HepG2, and U251 cell lines, solamargine upregulated the gene expression of Hep3B [91,92,103] and TNF-R1 (tumor necrosis factor receptor 1) [93]. Additionally, in the A549 and Hep3B cell lines, it also upregulated the gene expression of TNF-R2 (tumor necrosis factor receptor 2). Furthermore, solamargine significantly increased the protein expressions of Fas (also known as CD95) and other downstream signal proteins like FADD (Fas-associated death domain protein) and TRADD (tumor necrosis factor receptor type 1-associated death domain protein) in H441, H520, H661, B16F10, HT29, MCF-7, HBL-100, SK-BR-3, ZR-75-1, A549, HeLa, HepG2, MO59J, U343, and U251 cells. These findings suggest that solamargine activates the signal transduction of TNF receptors (TNFRs), leading to the activation of caspase-3 and -8 in A549, H69, HBL-100, ZR-75-1, SMMC-7721, H44, H520, H661, and SKBR-3 cells [104] This activation of caspases further supports the induction of apoptosis in the treated cancer cells. Solamargine can induce intrinsic apoptosis by regulating intrinsic apoptotic death mediators. In the case of solamargine treatment, it was observed that cytochrome c was released from the mitochondria into the cytosol in a dose-dependent manner in various cell lines, including A549, K562, H44, H520, H661, H69, HBL-100, ZR-75-1, SK-BR-3, and U2OS cells [93,96,98,103,104].

### 3.3. Solanine

A natural steroidal alkaloid—was initially identified in 1820 and subsequently recognized as prevalent in various plants, including the genus Solanum. It is categorized into α-solanine, β-solanine, and γ-solanine, with α-solanine exhibiting the highest concentration [110]. It acts as a natural defense mechanism for plants against herbivores and pests. While solanine is generally present in low concentrations in these plants, its level can increase under certain conditions, such as exposure to light and storage [111]. It has been observed to initiate the liberation of Ca^2+^ from the mitochondria, which consequently raises the cytoplasmic levels of Ca^2+^ within HepG2 cells. This outcome subsequently leads to a decrease in the potential of the mitochondrial membrane, ultimately instigating the process of apoptosis [112]. Additionally, in the HepG2 cell line, Ji et al. [113] demonstrated the induction of apoptosis as evidenced by the appearance of a sub-G0 apoptosis peak at various doses of α-solanine. Moreover, a reduction in the level of the antiapoptotic protein Bcl-2 was noted in a manner that correlated with the administered dose. In pancreatic cancer cells, α-solanine was discovered to induce an upregulation of p53 and Bax expression, concomitant with the downregulation of Bcl-2. Consequently, this event prompted the release of cytochrome c, thereby initiating the activation of the mitochondrial pathway for apoptosis. This decline in Bcl-2 and elevation in the Bax levels were likewise observed in cancerous tissue [113]. In a study by Mohsenikia et al. [114], it was demonstrated that α-solanine treatment in mice with breast cancer resulted in an increase in the proapoptotic Bax protein in breast cancer tissue. α-solanine has been found to exert another significant effect on cancer cells by inhibiting cell migration and invasion. This inhibition is achieved through the suppression of JNK, PI3K, and Akt phosphorylation, which subsequently leads to the downregulation of MMP-2 and MMP-9 expression. Moreover, α-solanine treatment has been shown to cause a reduction in the nuclear content of NF-κB in treated cells, further contributing to the inhibition of cancer metastasis [115]. In pancreatic cancer cells, Lv et al. [116] showed that α-solanine inhibits cell migration, invasion, and angiogenesis by reducing the levels of matrix metalloproteinases (MMPs) and vascular endothelial growth factor (VEGF). Additionally, α-solanine was found to modulate various signaling pathways, including JAK/STAT, Wnt/β-catenin, Akt/mTOR, and NF-κB/p65. These regulatory effects contribute to the inhibition of cell proliferation and the induction of apoptosis.

### 3.4. Apigenin

Formally classified as a flavone and recognized as 4′,5,7-trihydroxyflavone, stands out as one of the prevalent flavonoids found in a variety of plants. Its widespread presence is notably observed in plants within the Solanaceae and Asteraceae families, including those within the Artemisia genus [117]. In terms of biosynthesis, apigenin originates from the phenylpropanoid pathway and can be derived from both phenylalanine and tyrosine, two precursor molecules obtained through the shikimate pathway. Starting with phenylalanine, non-oxidative deamination leads to the formation of cinnamic acid, followed by oxidation at C-4. This intermediate is then transformed into p-coumaric acid. On the other hand, tyrosine undergoes direct deamination to produce p-coumaric acid. Upon CoA activation, p-coumarate combines with three malonyl-CoA units and undergoes aromatization through chalcone synthase to generate chalcone. Chalcone is subsequently isomerized by chalcone isomerase, resulting in the formation of naringenin. Finally, naringenin is oxidized into apigenin by flavanone synthase [118,119]. It manifests its impacts through the modulation of diverse kinase pathways, leading to the arrest of the cell cycle in the G2/M phase. Investigations have indicated that apigenin possesses the capability to impede cell proliferation and induce autophagy within HepG2 cells, exhibiting a reliance on both time and dose [120]. The autophagic mechanism in the HepG2 cells is facilitated through the suppression of the PI3K/Akt/mTOR pathway [121].

Similarly, other compounds from the Solanaceae follow varied mechanisms with different cellular backgrounds. Understanding these mechanisms can provide valuable insights for developing new cancer therapies that target specific pathways and exploit the vulnerabilities of cancer cells [122]. The cell cycle represents a pivotal mechanism governing cell proliferation. Notably, specific alkaloids sourced from the aerial parts of *Solanum nigrum* have exhibited antiproliferative attributes against gastric cancer cells. Some of these alkaloids, including β1-solasonine, solasonine, and solanigroside P, have shown the ability to induce apoptosis in gastric cancer cells. These alkaloids exert their antiproliferative effects by influencing the gene expression in cancer cells. They were found to increase Bax expression, which leads to the activation of apoptotic pathways, and decrease Bcl-2 expression, which is an anti-apoptotic protein that prevents cell death. By reducing Bcl-2 expression, the alkaloids promote the shift toward apoptosis, activating caspase-3, which triggers the breakdown of cellular components and leads to cell death [123].

## 4. Pharmacological Properties

Pharmacological properties refer to the specific effects and interactions of a drug or a compound with the biological systems of living organisms. These properties are responsible for the compound’s ability to exert therapeutic or toxic effects within the body. Pharmacological properties encompass a wide range of characteristics that determine how a substance behaves in the body and how it affects physiological processes (Table 2). Some common pharmacological properties for compounds isolated from different members of the Solanaceae family are listed as follows.

### 4.1. Anti-Cancerous Activity

The phytochemicals found in plants of the Solanaceae are known to possess anticancer properties. *S. surattense*, a medicinal plant, exhibits an anticancer efficacy attributed to the presence of specific phytochemicals, including solamargine, diosgenin, apigenin, and lupeol. These compounds are considered valuable sources of anticancer components and contribute to the plant’s potential anticancer activity [124]. In a study conducted by Kumar and Pandey, [125], it was reported that the fruit extract of *S. surattense* exhibited anticancer activity against human lung cancer cell lines (HOP-62) and leukemic cell lines (THP-1). The study demonstrated the potential anticancer efficacy of the fruit extract from *S. surattense* on these specific cell lines. Similarly, the methanolic extract of *S. nigrum* fruits was investigated for its inhibitory effect on the HeLa cell line (uterine cervix) [126]. The cell line viability was evaluated using the trypan blue dye exclusion technique. The cytotoxic impact of *S. nigrum* on the HeLa cells was gauged using both the MTT and SRB assays. The findings revealed that the methanolic extract of *S. nigrum* exhibited notable cytotoxic activity against the HeLa cell line across a concentration range spanning from 0.0196 mg/mL to 10 mg/mL, as ascertained using the SRB assay.

**Table 2 plants-13-00724-t002:** Structural–activity relationship of different bioactive compounds of the family Solanaceae.

Name	Structure	Plant Parts	Medicinal Use	Other Use	References
Solasodine	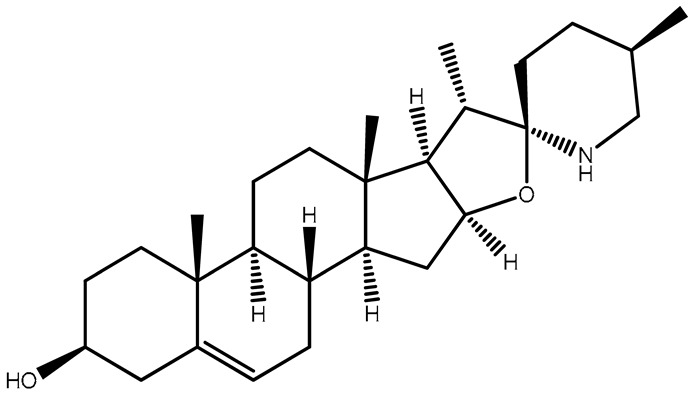	Immature berries	Suppresses cancer growth, antioxidant, cytotoxic, hepatoprotective, anti-inflammatory	Used in the production of tonics, creams, and lotions	[127]
Diosgenin		Roots	Antiproliferative and anti-inflammatory activities	Used in skincare products	[128]
Solanine	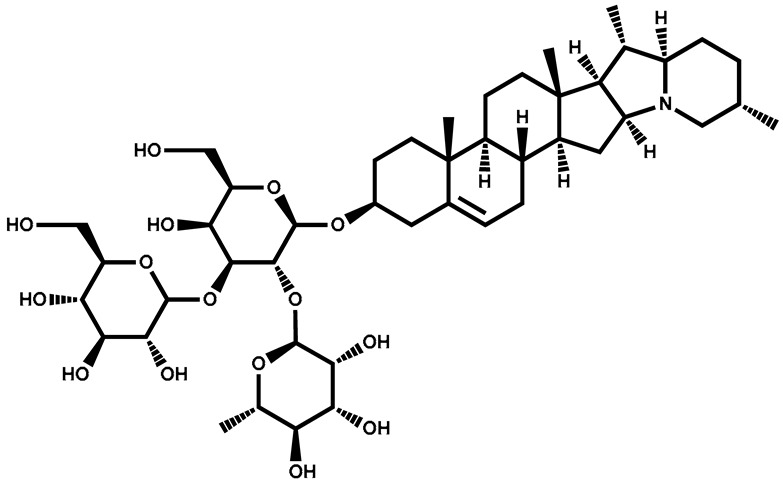	Berries	Anti-inflammatory activity on LPS-activated RAW 264.7 macrophages	Used as a pesticide and fungicide	[129]
Solasonine	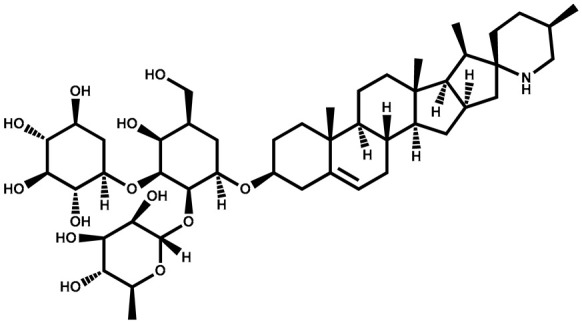	Berries	For production of contraceptives and steroidal anti-inflammatory drugs	Used as a pesticide	[130]
Solamargine	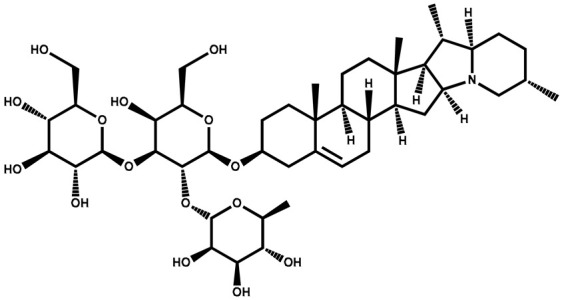	Berries	Antidiabetic, antifungal, antiparasitic, antibiotic, antimicrobial, and anti-cancerous properties. Cytotoxicity against skin tumors		[131]
Degalactotigonin	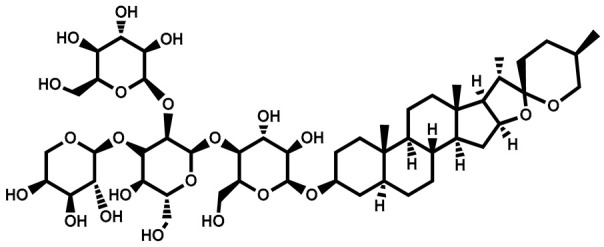	Whole plant extract	Suppressed the growth and metastasis of osteosarcoma	Used as a wood protectant, acaricide, and pesticide	[132]
Βeta-carotene	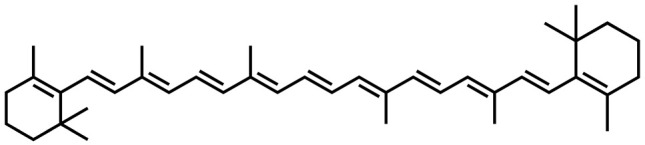	Leaves	Provides strong immune system and prevents cataracts	Used in confectionary, dairy, and packaged foods	[133]
Niacin	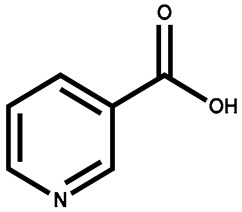	Leaves	Helps to keep the digestive and nervous systems healthy	Used in infant formulas, breakfast cereals, and energy drinks	[134]
Ascorbic acid	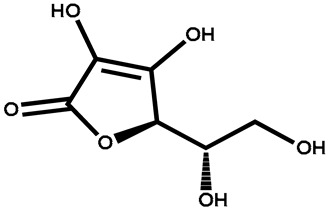	Leaves	To treat stomachache, jaundice, liver problems, and skin diseases	Used in hormone biosynthesis and as a cofactor	[135]
Citric acid	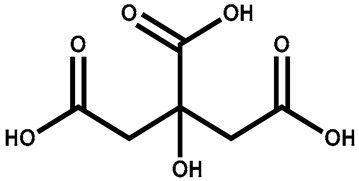	Leaves	Enhances Ca, P, and Mg absorption	Electroplating and leather tanning	[136]
Stearic acid	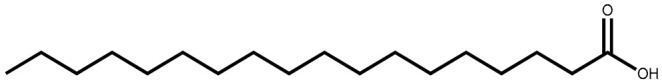	Seeds	Used as a tablet and capsule lubricant	Used in soaps, shaving creams, detergents, lotions, moisturizers, and candles	[137]
Palmitic acid	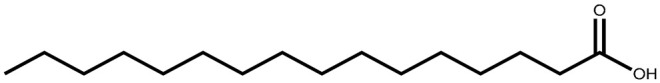	Seeds	Anti-inflammatory effect	Acts as an emollient, food additive, industrial mold release	[138]
Oleic acid	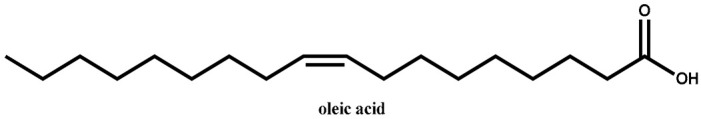	Seeds	Improves heart conditions by lowering cholesterol and reducing inflammation	Used as a lubricant, detergent, and surfactant	[139]
Linoleic acid	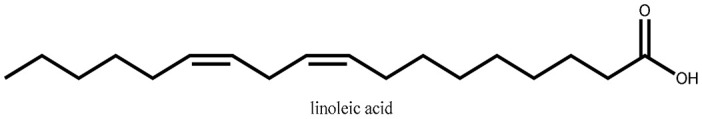	Seeds	To treat skin-related disorders	Used as moisturizer for skin, nails, and hair	[140]
Rutin	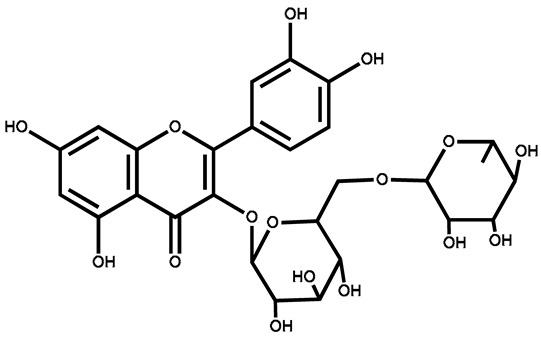	Berries	Antioxidant, cytoprotective, vasoprotective, anticarcinogenic, neuroprotective, and cardioprotective	Used as a colorant, antioxidant, and preservative	[141]
Protocatechuic acid	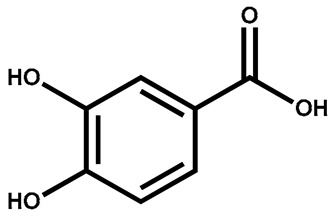	Leaves	Anti-inflammatory, antioxidant, estrogenic activity	Used to produce plastics and polymers	[142]
Gallic acid	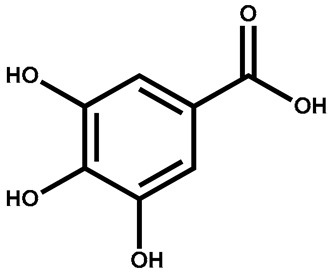	Leaves	Neuroprotective, anti-inflammatory, decreases myocardial infarction	Used in paper manufacturing and ink dyes	[143]
Naringenin	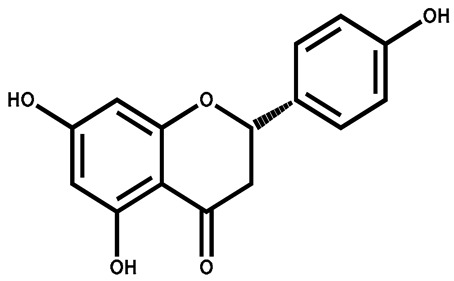	Leaves	Anti-HCV and cardioprotective	Used as flavoring agent in carbohydrate drinks	[144]
Isoquercitrin	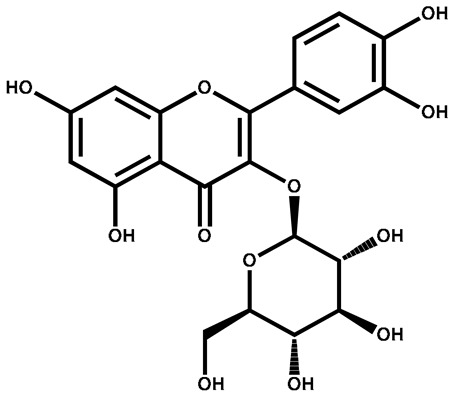	Berries/leaves	Antiviral, antioxidant, anti-inflammatory, stimulates mitochondrial biogenesis	Used in beverages	[145]

A series of indiosides (A–E) originating from *S. indicum* showcased a dose-dependent inhibitory influence on Bel-7402 cell proliferation, accompanied by the capability to induce cell death via the mitochondrial pathway. Furthermore, solavetivone-1, a compound present in *S. indicum*, was identified to possess cytotoxic properties against OVCAR-3 cells. These findings highlight the potential anticancer properties of these compounds derived from *S. indicum* [146]. The anticancer activity of *S. indicum* was investigated by testing the chloroform-soluble and insoluble fractions of the ethanolic extract obtained from the whole plant. In vitro techniques were employed to assess the anticancer potential against seven different cancer cell lines: HeLa, Hep-2 (laryngeal epidermoid), GBM8401/TSGH (glioma), H1477 (colon), Colo-205 (colon), KB (nasopharynx), and melanomas.

Using a dye exclusion assay (DEA) and 3-(4,5-dimethylthiazol-2-yl)-2,5-diphenyltetrazolium bromide (MTT) assays, it was determined that the purified components, dioscin and methyl protodioscin, exhibited more potent effects in terms of their anticancer activity. Moreover, in the polarographic reduction of oxygen (PRE) assay, dioscin, methyl protoprosapogenin A of dioscin, protodioscin, and methyl protodioscin exhibited cytotoxic impacts on cultured C6 glioma cells. Additionally, methyl protoprosapogenin A of dioscin, protodioscin, and methyl protodioscin demonstrated attributes of tumor suppression. Furthermore, dioscin, when administered at a concentration of 10 micrograms/mL, manifested an inhibitory influence on DNA synthesis within C6 glioma cells. These findings suggest the potential anticancer properties of these compounds, specifically in the context of C6 glioma cells [147]. Ma et al. [148] reported that several indiosides (A–E) extracted from *S. indicum* displayed a dose-dependent inhibitory effect on the proliferation of Bel7402 cells. Furthermore, these indiosides were found to induce apoptosis in the cells through the mitochondria-dependent pathway. Additionally, solavetivone-1, a specific component of *S. indicum*, demonstrated cytotoxicity against OVCAR-3 cells. These findings suggest the potential therapeutic value of these compounds in targeting cancer cells [146]. The cytotoxic influences of the chloroform-soluble and insoluble fractions from the ethanolic extract derived from the entire *S. indicum* plant were assessed via in vitro methodologies on seven distinct cancer cell lines. The tested cell lines encompassed HeLa, HA22T, Hep-2, GBM8401/TSGH, Colo-205, KB, and H1477. The refined compounds, notably dioscin and methyl protodioscin, exhibited more potent effects when subjected to evaluation using the DEA and MTT assays. These compounds demonstrated cytotoxicity in cultured C6 glioma cells.

Chiang et al. [147] reported that dioscin exhibited an inhibitory effect on DNA synthesis in C6 glioma cells at a concentration of 10 micrograms/mL. This indicates that dioscin holds promise in obstructing the replication and proliferation of C6 glioma cells. Moreover, the anticancer potential of the methanolic extract from the fruit was examined using the MTT cytotoxicity assay across diverse cancer cell lines. These encompassed prostate carcinoma (PC3 and DU145), colorectal carcinoma (HCT116), human non-small cell lung carcinoma (H1975), and malignant melanoma (A375). The evaluation of the extract’s cytotoxicity using the MTT assay indicates its potential as an anticancer agent against these specific cancer cell lines.

Gopalakrishna et al. [149] observed that the fruit extract of *S. indicum* exhibited its maximum cytotoxicity in the prostate carcinoma cell line, with an IC_50_ of 8.48 µg/mL in DU145 cells and 11.18 µg/mL in PC-3 cells. In H1975 cells, the extract exhibited cytotoxicity, reaching an IC_50_ value of 9.03 µg/mL. Likewise, in HCT116 cells, the extract displayed cytotoxic effects, resulting in an IC_50_ of 17.58 µg/mL. Furthermore, cytotoxicity was observed in A375 cells, yielding an IC_50_ of 27.94 µg/mL. These findings suggest the potential of the fruit extract in inhibiting the growth of these cancer cell lines. Additionally, Rahman et al. [150] found that the fresh fruit of *S. indicum* exhibited significant brine shrimp lethality, with an LC_50_ value of 4.42 ± 0.67 µg/mL. This indicates the potential bioactivity of the fruit extract against brine shrimp larvae, suggesting its possible cytotoxic properties.

The glycoalkaloids derived from the butanol extract of *S. villosum* fruit exhibited a toxic impact on the LIM-1863 human colon carcinoma cell line, contributing to cell death in these cancer cells. These naturally occurring compounds hold promise as valuable starting points for the exploration and development of prospective cancer therapeutics [151].

### 4.2. Antioxidant Activity

Oxidative stress is known to contribute to the development and progression of gastric ulcers, and antioxidants can help in mitigating this by reducing oxidative damage and promoting tissue healing. *S. torvum* has demonstrated significant antioxidant activity, which can help to reduce oxidative stress. Oxidative stress is associated with various health conditions, including diabetes, and can contribute to the development and progression of complications. The ability of *S. torvum* to scavenge reactive oxygen species (ROS) and mitigate oxidative stress makes it a potential natural remedy for managing oxidative-stress-related conditions like diabetes. By incorporating *S. torvum* into the diet or using its antioxidant components as a supplement, it may be possible to counteract the harmful effects of ROS and reduce the oxidative damage to cells and tissues. Indeed, in vitro studies have shown the promising antioxidant activity of this plant [152].

Antioxidants play a crucial role in preventing the oxidative damage caused by free radicals in the body, which is associated with various diseases. The restoration of antioxidant indicators such as glutathione, superoxide dismutase (SOD), glutathione reductase (GR), catalase (CAT), and lipid peroxidation (LPO) suggests that the extract possesses strong antioxidant properties. *S. surattense* has attracted attention due to its potential natural antioxidant properties. In a study by Meena et al. [153], it was discovered that methanolic and ethanolic extracts of *S. surattense* demonstrated notable antioxidant properties. This suggests that the plant contains compounds that can scavenge free radicals and protect against oxidative damage. The study suggests that the leaf extract of *S. surattense* possesses compounds that can potentially modulate the antioxidant defense system in the body [154]. By enhancing the levels of antioxidant enzymes such as SOD and GPx, the extract may help to counteract the increased oxidative stress caused by alloxan-induced diabetes [155]. By increasing the levels of these antioxidant enzymes, the leaf extract of *S. surattense* may contribute to reducing oxidative stress and protecting against cellular damage in alloxan-induced animal models.

A *S. nigrum* glycoprotein exhibited a radical scavenging activity that was dependent on the dosage. It demonstrated the ability to scavenge various radicals, including the hydroxyl radical (OH),1, 1-diphenyl-2-picrylhydrazyl (DPPH) radicals, and superoxide anions (O^2−^), in a dose-dependent manner [156]. The antioxidant activity of the methanolic extract obtained from *S. indicum* berries was evaluated using the in vitro DPPH radical scavenging assay. Additionally, both the aqueous and ethanolic extracts of *S. indicum* leaves demonstrated DPPH scavenging potential, indicating their ability to neutralize free radicals [157]. The extract exhibited the highest inhibition rate (70.007 ± 0.841%) at a concentration of 200 µg/mL [158]. In a distinct study, IC_50_ values were established for both ethanolic and aqueous extracts of berries through the utilization of the DPPH scavenging assay and the β-carotene/linoleate model system. Notably, the ethanolic extract displayed more pronounced effectiveness (IC_50_ 37.22 ± 1.3 µg/mL) in the β-carotene assay, while the aqueous extract demonstrated superior efficacy in the DPPH assay (IC_50_ 21.83 ± 0.84 µg/mL) [159]. Furthermore, studies have indicated that the antioxidant capacity of the fruit intensifies during the ripening process, as observed using the ferric-reducing antioxidant power (FRAP) test and Folin–Ciocalteau assay. This enhancement in antioxidant potential is likely attributed to a significant increase in the concentration of β-carotene in red berries. Nevertheless, specific compounds such as caffeoylquinic acids, caffeic acid, flavonol glycosides, and naringenin demonstrated an increase in concentration as the berries reached maturity. On the other hand, the levels of p-coumaric acid and feruloylquinic acids remained constant at all stages of ripening [160].

### 4.3. Hepatoprotective

The investigators explored the hepatoprotective potential of aqueous and methanolic extracts of *S. nigrum* in rats subjected to a 10-day consecutive regimen of carbon tetrachloride (CCl4) injections. Carbon tetrachloride is recognized for its hepatotoxic effects. The experimental procedure involved the initial administration of CCl4 to induce liver damage in the rats. Subsequently, the rats were treated with aqueous extract of *S. nigrum* at doses spanning from 250 to 500 mg/kg. The purpose was to assess whether the extract could provide protection against the liver damage caused by CCl4. According to the findings of the study, the aqueous extracts of *S. nigrum* demonstrated a hepatoprotective effect against CCl4-induced liver damage in the rats. The evidence for this hepatoprotective effect was observed according to several parameters such as decreased serum aspartate aminotransferase (AST), alanine aminotransferase (ALT), and alkaline phosphatase (ALP) activities: elevated levels of these liver enzymes in the blood are indicative of liver damage. The administration of the aqueous extract resulted in a decrease in the activities of these enzymes, suggesting a protective effect on the liver. The study found that the administration of the aqueous extract led to a decrease in bilirubin concentration, indicating improved liver function. In terms of mild histopathological lesions, histopathological examination involves the microscopic examination of tissue samples to assess any abnormal changes. In this study, the rats injected with CCl4 alone exhibited more severe histopathological lesions in the liver tissue. However, when the rats were treated with the aqueous extract of *S. nigrum*, the severity of these lesions was reduced. This suggests that the extract helped to mitigate the damage caused by CCl4. Similarly, the administration of methanolic extracts of *S. nigrum* at doses of 250 to 500 mg/kg exhibited hepatoprotective effects. Notably, the levels of serum AST, ALT, ALP, and bilirubin demonstrated a significant reduction in animals treated with *S. nigrum* methanolic extract when compared to an untreated control group [161].

A study conducted by Bhuvaneswari and Suguna, [162], investigated the hepatoprotective role of *S. indicum* extract in a rat model of hepatotoxicity induced by carbon tetrachloride (CCl4). The researchers administered *S. indicum* extract at a dose of 200 mg/kg to the rats with CCl4-induced hepatotoxicity. They evaluated various liver markers like ALT, AST, ALP, ACP, and LDH and biochemical parameters to assess the extent of the liver damage and the effects of the plant extract. Furthermore, the investigation encompassed an analysis of diverse biochemical indicators, encompassing total bilirubin, total protein, triglycerides, total cholesterol, and urea. The findings of the study indicated that the *S. indicum* extract significantly ameliorated the damage caused by CCl4. While specific details about the extent of improvement or the exact values of the parameters were not provided, the significant amelioration suggests a protective effect of the *S. indicum* extract on the liver.

### 4.4. Cardioprotective

One study focused on evaluating the cardioprotective potential of the methanolic extract sourced from *S. nigrum* berries. To achieve this, an in vitro model simulating global ischemia–reperfusion injury was utilized to gauge the impact of the extract on cardiac function. The extract was administered at doses of 2.5 and 5.0 mg/kg over six days per week, spanning a total duration of 30 days. The findings of the investigation strongly suggested that the methanolic extract displayed noteworthy cardioprotective effects against global ischemia–reperfusion injury within the in vitro model. This activity was observed in a dose-dependent manner, meaning that higher doses of the extract provided a stronger protective effect [163].

The study investigated the cardiotonic activity of the methanolic extract derived from the fruits of *S. indicum* using a frog heart model. The researchers administered the extract at concentrations of 5 and 10 mg/mL to assess its effects on the force of contraction and heart rate. The results of the study indicated that the methanolic extract exhibited marked cardiotonic activity in a dose-dependent manner. When administered at a concentration of 5 mg/mL, the extract induced a modest enhancement in the force of contraction; however, no substantial alterations in heart rate were detected. Conversely, at a higher concentration of 10 mg/mL, the extract elicited a noteworthy escalation in the force of contraction, accompanied by a slight elevation in heart rate. Importantly, the study found that the methanolic extract of *S. indicum* exhibited a wide therapeutic index, meaning that it had a significant cardiotonic effect without showing any signs of cardiac toxicity at the higher tested doses, up to 5 mg/mL. These findings suggest that the methanolic extract of *S. indicum* fruits possesses cardiotonic activity, meaning it can enhance the force of contraction in frogs’ hearts. The extract demonstrated a dose-dependent response, with higher concentrations resulting in more pronounced effects on cardiac function. Importantly, the extract exhibited a wide therapeutic index, indicating a relatively safe profile without causing cardiac toxicity at the tested higher doses [164].

### 4.5. Nephroprotective

In a study conducted by Waghulde [165], the nephroprotective effects of *S. torvum* fruits against doxorubicin (DOX)-induced nephrotoxicity were demonstrated in rats. DOX is recognized for its ability to cause harm to kidney cells in both rat and human subjects by triggering an excessive production of free radicals of the semiquinone type [166,167]. The flavonoids present in *S. torvum* have antioxidant and nephroprotective properties, as reported by Ching et al. [168]. These flavonoids are also capable of chelating free iron, which contributes to a reduction in DOX-induced toxicity in the kidneys. This suggests that the antioxidant and iron-chelating properties of *S. torvum* flavonoids play a role in mitigating the nephrotoxic effects induced by DOX [169]. Mohan et al. [170] conducted a histopathological analysis of the kidneys and observed that *S. torvum* reversed the structural damage caused by DOX, including tubular necrosis, renal lesions, and glomerular congestion. This suggests that *S. torvum* can mitigate the detrimental effects of DOX on the kidneys and restore their normal structure. Loganayaki et al. [171] also mentioned the nephroprotective action of phenolic compounds extracted from different parts of *S. torvum*. These phenolic compounds may possess properties that helps in protecting the kidneys from damage and maintain their normal function.

### 4.6. Hypertensive and Anti-Thrombotic Activity 

Aqueous extract derived from dried fruits of *S. torvum* has demonstrated the ability to lower blood pressure. This effect may be attributed to a decrease in the sensitivity of vasorelaxant agents and an increase in hypersensitivity to contractile factors. In laboratory experiments conducted *in vitro*, the extract exhibited potent vasocontractile activity by activating both the Alpha 1-adrenergic pathway and calcium reflux [172]. Considering these observations, the potential utility of *S. torvum* in managing severe hypotension becomes apparent, especially in scenarios stemming from autonomic dysfunction that necessitate the application of vasopressor agents.

The potential anti-thrombotic impact of the aqueous extract derived from *S. torvum* was additionally examined using isolated rat platelets. Notably, the intravenous administration of both aqueous and methanolic extracts led to a noteworthy decrease in arterial blood pressure. This observed anti-thrombotic aggregation effect of *S. torvum* could hold significance for its potential cardiovascular implications in conditions such as arterial hypertension and hemostatic disorders [173]. The impact of a standardized ethanolic extract originating from *S. indicum* fruit (containing > 0.15 percent chlorogenic acids) on blood pressure was investigated in both normotensive and hypertension-induced (N(W)-nitro-L-arginine methylester (L-NAME) treated) rats. In normotensive rats, the administration of the extract (30 mg/kg) for a duration of four weeks did not exhibit any influence on blood pressure. However, after L-NAME administration, the extract effectively prevented the onset of hypertension in the animals [174]. The hypotensive effects of a standardized ethanolic extract sourced from *S. indicum* fruit (containing > 0.15% chlorogenic acids) were assessed in both normotensive and hypertensive (N(W)-nitro-L-arginine methyl ester (L-NAME)-treated) rats. The administration of the extract at a dose of 30 mg/kg for a duration of four weeks did not induce hypotensive effects in normotensive rats. However, it effectively averted the development of hypertension in the animals following L-NAME administration [175]. Similarly, an ethanolic extract of *S. villosum* exhibited mild antihypertensive activity in experimental rats. However, the study did not provide detailed information on the specific mechanisms of action or active compounds responsible for this effect [176].

### 4.7. Anti-Ulcerogenic

The presence of flavonoids, sterols, and triterpenes in *S. torvum* suggests that these compounds may contribute to its anti-ulcer properties. *S. torvum* is known to strengthen the mucosal barrier by promoting the production of mucus and bicarbonate. It may also reduce the volume of gastric acid secretion or neutralize gastric activity, thereby potentially preventing the development or progression of gastric ulcers [177]. Rats were subjected to different stress-inducing methods including cold restraint stress, indomethacin administration, pyloric ligation, ethanol ingestion, and acetic acid exposure, in order to induce stress ulcers in the experimental model. The extract from *S. torvum* fruits demonstrated a significant inhibition of the gastric lesions induced by these stressors, with the percentages of inhibition ranging from 70.6% to 80.1%. The potency of the extract was equal to or higher than that of omeprazole, a known anti-ulcer medication. The administered extract demonstrated a decrease in gastric secretory volume, acidity, and pepsin secretion in the rats afflicted with ulcers. Furthermore, a 7-day administration of the extract expedited the healing of the ulcers induced by acetic acid. To investigate the anti-secretory action of the extract, enzymatic studies were conducted on H^+^/K^+^ ATPase activity. The results showed that the *S. torvum* fruit extract significantly inhibited H^+^/K^+^ ATPase activity, which plays a role in gastric acid secretion. Additionally, the extract reduced the gastrin secretion in an ethanol-induced ulcer model, indicating its potential in modulating gastric secretory functions. These findings suggest that the extract from *S. torvum* fruits possesses anti-ulcer properties, inhibits gastric acid secretion, and accelerates ulcer healing [178].

The potential anti-ulcerogenic effects of the methanolic extract derived from the fruit *of S. indicum* were investigated in rats with induced ulceration caused by aspirin and ethanol. The administration of the extract at a dose of 750 mg/kg demonstrated significant effects. One of the key findings was that the extract protected the stomach mucosa from the damaging effects of aspirin and ethanol. This indicates its potential as a gastroprotective agent by preventing ulcer formation. Additionally, the extract exhibited the added benefit of promoting ulcer repair. This suggests that it can aid in the healing process of existing ulcers, which is crucial for restoring the integrity of the gastric mucosa. The observed anti-ulcerogenic effects of the extract are likely attributed to its antioxidant capacity. These effects are likely mediated through its antioxidant capacity, which helps restore the balance of oxidative stress in the stomach [179]. The objective of the rat study was to assess the choleretic activity of a suspension containing the fruit of *S. indicum*. The rats were initially anesthetized with intraperitoneal sodium pentobarbital. Then, they were exposed to bile duct cannulation, where a catheter was inserted to collect bile. Before administering the plant solution, bile was collected for one hour to establish the baseline levels. After this initial collection period, the rats were intraduodenally administered the plant solution at a dosage of 500 mg/kg. Following the administration of *S. indicum*, the researchers observed a significant increase in bile flow. Specifically, there was a 31 percent increase in bile flow compared to the baseline levels [180]. This finding indicates that the fruit of *S. indicum* exhibited choleretic activity via stimulation of the production and secretion of bile from the liver into the bile ducts.

### 4.8. Analgesic and Anti-Inflammatory Activity

The peripheral analgesic activity of aqueous extract obtained from *S. torvum* leaves was examined, revealing its potential in providing relief from pain. Its reported analgesic properties have been attributed to the inhibition of prostaglandin synthesis. By inhibiting their synthesis, the aqueous extract of *S. torvum* leaves may reduce pain perception and provide relief from pain. However, the dosage, mode of administration, and potential side effects should be considered before applying this plant extract for pain management [181].

The documented anti-inflammatory potential of *S. torvum* extract suggests its capacity to potentially modulate the later stages of the inflammatory response by inhibiting cyclooxygenase, an enzyme pivotal in prostaglandin synthesis, known to mediate inflammation. Correspondingly, an examination of the anti-inflammatory attributes of the methanolic extract from entire *S. nigrum* plants was conducted using animal models. The study revealed that the methanolic extract, administered at 100 mg/kg and 200 mg/kg body weight concentrations, exhibited significant and dose-dependent anti-inflammatory effects in rats afflicted with hind paw edema induced by carrageenin and egg white [182]. These findings suggest that both *S. torvum* and *S. nigrum* extracts possess anti-inflammatory properties, which can potentially be attributed to their ability to modulate inflammatory processes and inhibit the production of inflammatory mediators.

The anti-inflammatory potential of ethanolic extracts from *S. nigrum* was assessed utilizing the rat paw edema model as induced by carrageenan. Oral administration of varying doses—100 mg/kg, 250 mg/kg, and 500 mg/kg—was conducted. Notably, the study highlighted a significant anti-inflammatory effect (*p* < 0.001) of the 500 mg/kg dose of the extract in comparison to the reference drug, diclofenac sodium (50 mg/kg). Similarly, the impact of methanolic extracts sourced from *S. nigrum* berries was investigated concerning carrageenan-induced paw edema. The results indicated a notable reduction in hind paw edema due to the methanolic extract. Particularly, the anti-inflammatory potential was significant at a dose of 375 mg/kg body weight. These findings suggest that both ethanolic and methanolic extracts of *S. nigrum* possess anti-inflammatory properties [183,184].

### 4.9. Antibacterial and Antiviral Activity

The ethanolic leaf extract of *S. surattense* has been found to possess potential antimicrobial activity against various pathogens. This extract has shown activity against bacteria such as *Vibrio cholera*, *Pseudomonas aeruginosa*, *Staphylococcus aureus*, *Streptococcus species*, *Escherichia coli*, *Salmonella typhi*, and *Shigella dysenteriae*. These findings suggest that the ethanolic leaf extract of *S. surattense* may have the ability to inhibit the growth of or kill these microorganisms, indicating its potential as an antimicrobial agent [10]. Ethanol and methanol extracts of *S. surattense* have been found to possess strong antibacterial activity against *Pseudomonas aeruginosa* [185]. Additionally, the fruit extract of *S. surattense* has shown potential inhibition of the growth of bacteria such as *Escherichia coli*, *Salmonella typhi*, *Micrococcus luteus*, *Staphylococcus aureus*, *Pasteurella multifida*, and *Vibrio cholera*. These findings suggest that extracts of *S. surattense* may contain bioactive compounds with antibacterial properties, making them effective against a range of bacterial pathogens [186]. *S. surattense* extract has been studied for its antifungal effectiveness against several fungi, including *Aspergillus niger*, *A. flavus*, *A. fumigatus*, and *Trichoderma viride* [10]. Mahmood et al. [187] conducted a targeted investigation into the antifungal properties of *S. surattense*, focusing on its impact on the growth of *A. fumigatus* and *A. niger*. The study revealed noteworthy antifungal efficacy within the methanolic extract derived from *S. surattense* seeds, particularly exhibiting substantial activity against *Rhizopus oryzae* as well as *A. fumigatus*. These findings suggest that the methanolic seed extract of *S. surattense* may possess compounds that can inhibit the growth of certain fungal pathogens, making it a potential antifungal agent [188].

Research has also aimed to assess the antibacterial potential of methanol and aqueous extracts sourced from *S. nigrum* leaves. Employing the disc diffusion technique, the extracts were subjected to screening against two Gram-negative bacterial strains, *Xanthomonas campestris* (a plant pathogen) and *Aeromonas hydrophila* (an animal pathogen). The findings unveiled the notable antibacterial effectiveness of the methanol extracts from all plant samples, manifesting significant activity against both tested bacterial strains. Clear zones of inhibition were observed for the methanol extracts of *S. nigrum*, indicating their potential antibacterial properties against the tested microorganisms [189]. The ethanolic extract of *S. indicum* leaves exhibited antibacterial activity against *Pseudomonas* spp., *Corynebacterium diptheriae*, and *Salmonella typhimorium*, as reported by Gavimath et al. [190]. It has also been found that the ethanolic extract of *S. indicum* leaves showed antibacterial activity against *Bacillus cereus*, *Staphylococcus aureus*, and *Escherichia coli*. Additionally, the chloroform extract, acetone extract, and ethanol extract of *S. indicum* demonstrated antibacterial activity against Pseudomonas species [191]. The fruits of *S. indicum* have been found to possess antibacterial activity. Both aqueous and ethanolic extracts of the fruits showed effectiveness against *Escherichia coli*, *Listeria innocua*, *Staphylococcus aureus,* and *Pseudomonas aeruginosa* strains. It was observed that the ethanolic extract exhibited better activity compared to the aqueous extract in terms of its antibacterial effectiveness [192]. In the study, the concentration-dependent inhibitory influence of the aqueous fraction from the ethanolic extract of *S. indicum* berries was demonstrated against diverse Pseudomonas strains, notably encompassing *Pseudomonas fluorescens*, *Pseudomonas aeruginosa*, and *Pseudomonas syringae.* The tested aqueous fraction was reported to contain flavonoids, carotenoids, and saponins, which could contribute to its antibacterial properties [192].

Ethanol, methanol, and water are commonly used as polar extraction solvents for the plant parts of *S. incanum*, and they have been found to yield phytochemicals with antibacterial and antifungal properties [193,194,195]. In studies, the antibacterial effects of ethanol and aqueous crude extracts derived from the leaves, fruits, and stems of *S. incanum* were evaluated against established strains of both Gram-positive and Gram-negative bacteria. The crude extracts from *S. incanum* displayed diverse degrees of growth restraint against the bacterial strains under investigation. While certain extracts demonstrated no discernible inhibition, others exhibited significant inhibitory effects. Specifically, the aqueous stem extract did not show any growth inhibition. However, the ethanol extracts of the leaves, fruits, and stems demonstrated significant growth inhibition against the tested bacterial strains. Among these, the ethanol and aqueous leaf extracts were found to be particularly effective in inhibiting bacterial growth [196,197]. Assessment of the antibacterial efficacy of *S. villosum* leaf extracts was conducted against a pair of Gram-positive bacteria, as well as two Gram-negative bacteria. It was observed that all strains displayed susceptibility to its aqueous, n-hexane, and ethanol extracts. Notably, the organic extracts exhibited superior effectiveness compared to the aqueous counterparts. The leaf extracts of *S. villosum*, encompassing aqueous, n-hexane, and ethanol compositions, demonstrated substantial effectiveness against the entire spectrum of bacterial strains examined, thus indicating their potential utility against pathogenic microorganisms affecting humans [198].

The methanolic extract of *S. torvum* fruits was examined for its antiviral activity against herpes simplex virus type 1 (HSV-1). During the extraction process, a new C4-sulfated isoflavonoid called torvanol A, a steroidal glycoside named torvoside A, and torvoside H were derived from the extract. The researchers found that these compounds exhibited antiviral activity against HSV-1. This suggests that the methanolic extract of *S. torvum* fruits, containing torvoside A, torvanolA, and torvoside H, has the potential to suppress the replication or activity of HSV-1 [199].

### 4.10. Anthelmintic Activity

Gunaselvi et al. [200] reported that aqueous extracts of *S. surattense* fruit exhibit anthelmintic activities. The findings suggest that aqueous extracts of *S. surattense* fruit powder possess properties that may be effective against helminth infections. A study by Priya et al. [201] indicates that various extracts of the whole plant of *S. surattense*, including aqueous, hydroethanolic, and ethanolic extracts, exhibit anthelmintic activity. Furthermore, the anthelmintic potential of the butanol and aqueous fractions obtained from the methanolic extract of *S. indicum* fruits was evaluated using the *C. elegans* bioassay. This assay involves assessing the percentage of dead nematodes after a 24 h incubation period. The fractions eluted from DEAE cellulose showed anthelmintic activity, and this activity was observed at four separate peaks based on the *C. elegans* assay. The fractions obtained from the methanolic extract of *S. indicum* fruits, specifically the butanol and aqueous fractions, exhibited eluted peaks at NaCl concentrations of 0.1, 0.28, 0.48, and 0.85 M. These eluted peaks correspond to distinct compounds possessing anthelmintic properties. Notably, the peak-associated mean death percentages were 37%, 53%, 59%, and 61%, respectively, in comparison to the negative control. These findings suggest that the *S. indicum* fruit contains at least four different anthelmintic compounds, each contributing to the observed anthelmintic activity. Further characterization and identification of these compounds would be necessary to determine their specific structures and mechanisms of action. This information could potentially contribute to the development of new anthelmintic treatments or the isolation of active compounds for further investigation [202]. A methanolic extract of *S. indicum* berries at a concentration of 100 mg/mL demonstrated paralyzing effects on the Indian earthworm (*Pheretima posthuma*) within an average time of 9.16 ± 0.12 s. Additionally, helminths died within an average time of 17.71 ± 0.21 s after exposure to the extract. These results indicate the potential anthelmintic activity of the methanolic extract of *S. indicum* berries against the Indian earthworm. The observed paralysis and subsequent death of the helminth suggest the presence of bioactive compounds in the extract that could be responsible for these effects [158]. These findings suggest that both *S. surattense* and *S. indicum* possess compounds with potential anthelmintic properties. Nevertheless, additional investigations are warranted to ascertain the precise active constituents accountable for the noted anthelmintic effects and to explore their potential application to the management of helminth infections.

### 4.11. Antiplasmodial Activity

Research conducted by Zirihi et al. [203] was centered on the assessment of the ethanolic fruit extract of *S. indicum* for its antiplasmodial potential against the chloroquine-resistant FcB1 strain of *Plasmodium falciparum*, the causative agent of malaria. Furthermore, the extract’s impact on human MRC-5 and rat L-6 cell lines was examined to gauge its cytotoxicity. The study’s outcomes revealed the noteworthy antimalarial efficacy of the ethanolic fruit extract against the *Plasmodium falciparum* strain resistant to chloroquine. The IC_50_ value, which represents the concentration required to inhibit the growth of the parasite by 50%, was determined to be 41.3 ± 7.0 g/mL for the extract. Furthermore, cytotoxicity tests were conducted on human MRC-5 and rat L-6 cell lines to assess the potential toxicity of the extract on human and rat cells. The IC_50_ value, which represents the concentration required to cause a 50% reduction in cell viability, was found to be greater than 50 g/mL for both cell lines. This suggests that the extract exhibited low cytotoxicity against these cell lines at the concentrations tested. In summary, the ethanolic fruit extract demonstrated significant antiplasmodial activity against the chloroquine-resistant FcB1 strain of *Plasmodium falciparum*. The extract exhibited an IC_50_ value of 41.3 ± 7.0 g/mL, indicating its potency in inhibiting the growth of the parasite. Additionally, the extract showed low cytotoxicity on the human MRC-5 and rat L-6 cell lines, with IC_50_ values greater than 50 g/mL. These findings highlight the potential of the ethanolic fruit extract as a source of antimalarial compounds for further exploration and development [193]. The crude extracts obtained using methanol, petroleum ether, chloroform, ethyl acetate, and aqueous solvents from this plant demonstrated considerable and suitably specific antiprotozoal efficacy against *Trypanosoma brucei*, *T. cruzi*, *Plasmodium falciparum*, and *Leishmania infantum* [204].

### 4.12. Antimalarial Activity

The utilization of *S. surattense*, both in in vivo and in vitro contexts, has exhibited notable antimalarial effects without any associated toxicity. Notably, dichloromethane extract from *S. surattense* has displayed robust anti-plasmodial efficacy, underscoring its potential as a promising intervention for malaria treatment. Furthermore, Zihiri et al., 2005 [203], reported that an ethyl acetate extract derived from the aerial parts of *S. surattense* exhibited efficacy against the larvae of *Plasmodium falciparum*, the parasite responsible for causing malaria. This suggests that specific compounds present in the extract possess antimalarial properties and could potentially be developed into antimalarial drugs. These findings highlight the promising potential of *S. surattense* as a natural source for discovering new antimalarial compounds.

## 5. Conclusions and Future Perspectives

Traditional herbal medicines, often used in curing different human ailments, have gained significant momentum for being vital in the maintenance of the health and wellbeing of humans. Being rich in phytochemical compounds, their use in medical practice owing to their pharmacological attributes is often considered to have been a living tradition since time immemorial. The existing literature on plants with medicinal properties was used to describe their effectiveness against particular or multiple human diseases. To be precise, there was previously no in-depth study performed on the clinical efficacy of Solanaceae family members in terms of the plant parts used, phytochemical production, and correlation made between phytochemicals in line with their possible clinical applications. This paper aimed to describe the common but prominent members of the Solanaceae family grown across different sub-continents with special mentions of the parts used and metabolites screened for the evaluation of their therapeutic potential in diverse clinical applications, ranging from curing different diseases to their use in the treatment of different cancers. Of the plants studied, special stress has been placed on the production of different secondary metabolites, highlighting their structural diversity and elaborating on the mechanistic insights to resolve their modi operandi against different cellular backgrounds. The health benefits are covered in terms of their broad-spectrum use as anti-cancerous, antioxidant, hepato-, cardio- and nephroprotective, antihypertensive, anti-ulcerogenic, analgesic and anti-inflammatory agents, as well as being used as antibacterial, antiviral, anthelmintic, antiplasmodial, and antimalarial treatments. Despite their remarkable pharmacological properties, a gap exists between the therapeutic potential of the bioactive moieties and the clinical outcomes, limited by fewer studies having been performed on the structural–activity relationships for the improvement of their properties, not to mention the need to perform safety and potency checks of metabolites with drug-like properties. An in-depth understanding of the structural–activity relationship will help in enhancing their role within the system for use in the prevention and treatment of different ailments. Additionally, it will help in outlining a direction for future investigations to confirm their therapeutic properties and reap the benefit in terms of products for human welfare.

## Figures and Tables

**Figure 1 plants-13-00724-f001:**
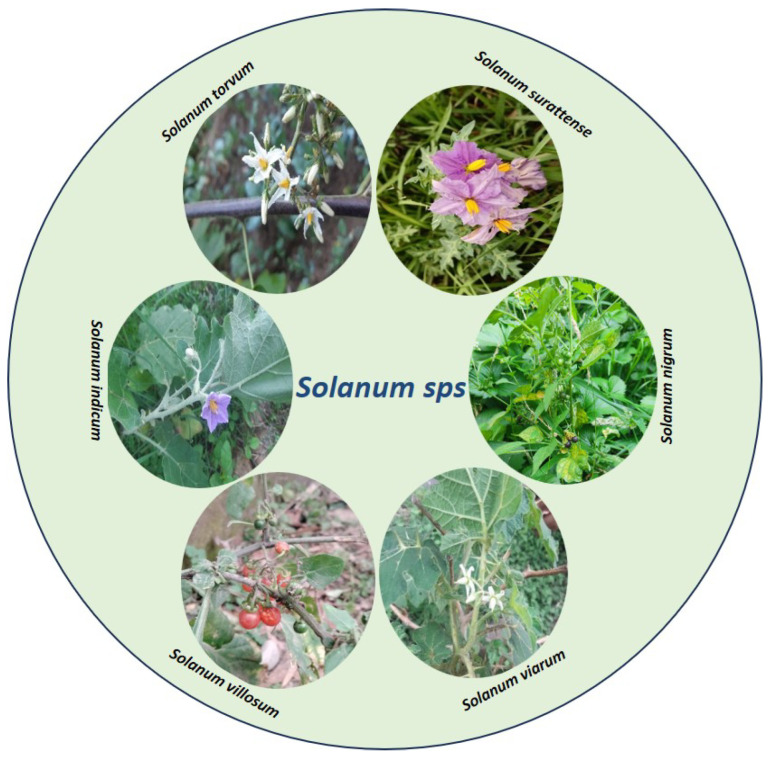
Representation of different members of the genus Solanum.

**Figure 2 plants-13-00724-f002:**
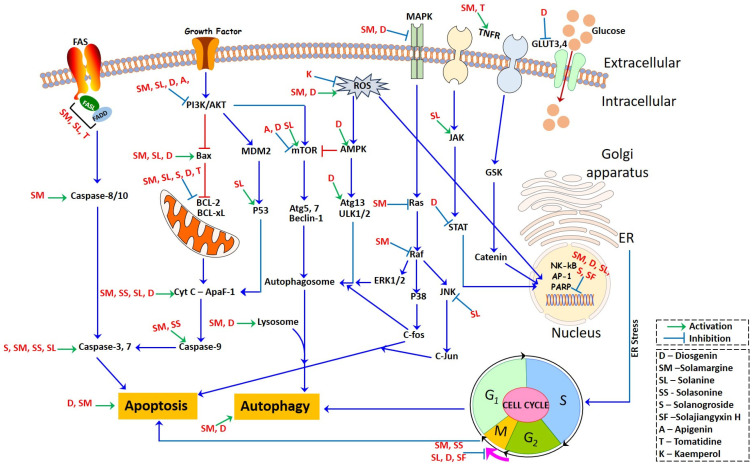
Molecular machinery of bioactive moieties governing different signaling pathways.

**Table 1 plants-13-00724-t001:** Comparison of the morphological parameters across different species of Solanaceae.

	*S. viarum*	*S. surattense*	*S. nigrum*	*S. villosum*	*S. incanum*	*S. indicum*	*S. torvum*
**Habit**	Shrub; perennial	Herb; perennial	Herb; annual	Herb; perennial	Perennial; bushy herb or shrub	Herb or small shrub	Shrub; perennial
**Root system**	Tap root	Tap root	Tap root	Tap root	Tap root	Tap root	Tap root
**Stem**	Stem densely yellow, hirsute, and with straight patent prickles. The hooked prickles are shorter than straight ones	Stem profusely branched somewhat zig zag, young branches with dense satellite and tomentose hairs, prickles compressed straight, glabrous, and shiny	Stem often angular, sparsely pubescent.	Stems decumbent, terete to ridged, green to purple; new growth densely pubescent with simple, uniseriate, translucent, and glandular trichomes	Stem is erect and covered with spines, green to purple in color	Upright, spiky and sturdy stem	Stems erect, branched and armed prickles sparse.
**Leaves**	Ovate–triangular, sinuate-lobed, lobes sub-obtuse or subacute, prickles straight, scattered; trichomes viscid, dense, straight or stellate	Ovate–elliptic, deeply lobed base attenuate, veins and margins with spines. Ovate or elliptic sinuate or sub-pinnatifid glabrescent, very prickly	Ovate or oblong, sinuate-toothed or lobed and glabrous	Leaves rhombic to ovate –lanceolate, margins are entire to sinuate–dentate	Leaves are simple, alternate, ovate, or elliptic	Leaves are broad, oblong in form, sub-entire with a few triangular—oval lobes, sparsely prickly and hairy on both sides; the base is cordate or truncate	Elliptic, oval, ovate or oblong, entire or irregularly lobed, acute or obtuse, highly variable; prickles straight or a few hooked; trichomes dense, stellate
**Inflorescence**	Extraaxillary raceme	Axillary cymose	Extraaxillary raceme	Inflorescence is simple, umbellate to slightly solitary cymes	Sometimes solitary or in clusters	Racemose extra axillary cymes, with short, stellate, hairy peduncles	Many flowered corymbose cymes
**Flower**	Sepals lanceolate and hirsute and corolla nearly glabrous. Corolla white, deeply lobed, petals recurved, anthers same length. Has both male and bisexual flowers and a style longer than its anthers	Bluish pink extra-axillary racemes, calyx lobes free, obovate, prickly acuminate. Corolla ovate–triangular	Calyx cup-shaped, white corolla with yellow anthers, lobes ovate–oblong, pubescent abaxially, ciliate spreading. Sepals ovate–oblong, calyx teeth small and obtuse. Corolla nearly glabrous	Calyces 1.2–2.2 mm long, slightly a crescent, deflexed or adhering to base of mature berry	Yellow or white calyx is fused, the purple corolla regular, bell- or wheel-shaped with five stamens	Corolla pale purple, covered with darker purple stellate hair on the outside; the stamens are attached to the corolla, with short filaments and anthers that are large	Calyx lanceolate, sparingly hairy; corolla lobes triangular and pubescent Corolla white, shallowly lobed, star-shaped; anthers same length
**Fruit**	Round, pale green with dark green veins, turning dull yellow at maturity.	Globose, yellow when ripe	Globose, dull black in color	Berries are usually longer than wide, orange, 6–10 mm broad,	The fruit is fleshy, less than 3 cm in diameter. Fruits are yellow then turn black	Globose, approximately 0.8 cm in diameter and dark yellow when ripe; glabrous	Round, green turning yellow then orange or brown at maturity. Seeds compressed, light brown
**Seed**	Seeds are flattened, discoid, and brown in color	Seeds discoid, smooth to faintly reticulate	Many discoid yellow seeds	Seeds are flattened and teardrop-shaped with a subapical hilum	Many pale brown seeds	Seeds are 0.4 cm in diameter and have minute pits	Seeds discoid
**References**	[39,42]	[43]	[44]	[33]	[11,41]	[45]	[46]

## Data Availability

Not applicable.

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
