# Peer review of "Unleashed Treasures of Solanaceae: Mechanistic Insights into Phytochemicals with Therapeutic Potential for Combatting Human Diseases"

_plants, 2024, doi:10.3390/plants13050724_

Round 1

Reviewer 1 Report

Comments and Suggestions for Authors

Comments on the Quality of English Language

The English language must be improved and simplified, even by avoiding repetitions in sentence.

Author Response

Authors would like to thank reviewer for the suggestions. Authors have addressed all the queries as mentioned below:

Query 1: Authors must fully revise the abstract. Here, the English language must be improved and simplified, even by avoiding repetitions in sentences (to make an example “revealed them” or “serve the purpose”, etc) to make the reading easier and more fluent. 

Response: The abstract has been rewritten as per the suggestion of the reviewer.

Query 2: Authors must take care to eliminate the high number of repetitions in the Introduction section (i.e., “members”, “distinct”, “hold”, “placed on” and so on) as well as to carefully check the whole manuscript because of the presence of different typos (such as “antitumorous” of line 50).

Response: The repetition has been minimized in the revised manuscript.
Query 3: Authors should move Table 1 to the end of the paragraph “Distribution and Morphology” and, here, they could also add a new Figure depicting the Solanaceae species that are described in the Table 1.

Response: The table has been shifted and a new figure has been added as per the suggestion.
Query 4: Authors must explain all acronyms when they appear in the text for the first time.

Response: Done.
Query 5: Authors should introduce a summary image after the description of diosgenin-regulated pathways (see paragraph 3). Similarly, they should do the same for the other metabolites (i.e., solamargine, solanine and apigenin). At the same time, they should remove Figure 1, since it reports information on all metabolites in a confusing and not-so-capturing way for the reader. 

Response: The figure has been drawn to summarize all the information about the involvement of bioactive compounds in different signalling pathways. If the figure is split, the figures will almost appear the same and as repetition to each other. 
Query 6: Authors must correct the typo “diosgenin” into “solamargine” in line 347.

Response: The correction has been done.
Query 7: Authors must use the term “dose” only when they refer to animal or human experimental models.

Response: The use of the term dose has been taken care of.
Query 8: Authors must correct the title of line 720 with “Anti-hypertensive effect and anti-platelet aggregation activity”.

Response: Changes has been done.
Query 9: Authors must number each paragraph or sub-paragraph in the text.

Response: The line number is automatic and cannot be manualized. 
Query 10: In the Conclusion section, Authors emphasize the clinical efficacy or the potential clinical applications of members of the Solanaceae family, although most of the discussed studies were carried out in vitro or in vivo. Therefore, Authors should be more consistent with the cited studies in the Conclusion.

Response: Necessary changes have been done to avoid any confusion.
Query 11: Authors should reduce old citations and update them with the latest ones

Response: There are not too many studies done on the subject. The references have been updated for the recent and available ones. 

Reviewer 2 Report

Comments and Suggestions for Authors

The authors have presented a literature review on the bioactive properties of plants from the Solanaceae family. The work is generally well done, interesting, and relevant to the area studied.

However, small details need fine-tuning.

   1) There is too much overlap between what is described in tableTable 1 and the text in linelines 82-173. This could be simplified.

2) A short paragraph should be written to explain the emphasis on the molecules diosgenin, solamargin, solanine and apigenin.

 Other minor issues:.

Line 180- define SMs if they are secondary metabolites, the extension for is used in line 181 so the abbreviation is not necessary

Fig. 1. -You should cite the source if it is not an original graphic, and the state the program used to create the figure.

Authors should clarify the origin of secondary metabolites. For example, there are 4 sections on diosgenin, but it is not stated that diosgenin is present in every plants. If not, it should be included the specie. The same applies to solamargine and solanine etc

Line 528   “from 10 mg/ml to 0.0196 mg/ml” the smallest number should be the first. Or do you mean 10 µg/ml?

Table 2- : Have these compounds been identified in all plants of the Solanaceae family? If not, the plant should be added to the table.

 In line 628-Ec50 the units are missing.

Line 697- 5 gm/ml ?  do you mean mg?

Author Response

Authors are thankful to the reviewer for the suggestions. Authors have incorportaed all the changes as mentioned below:

 1) There is too much overlap between what is described in Table 1 and the text in lines 82-173. This could be simplified.

Response: Tabulated information gives the overview and text information carries all the details pertaining to it. Authors have tried their best to simply the information. 

2) A short paragraph should be written to explain the emphasis on the molecules diosgenin, solamargin, solanine and apigenin.

Response: As suggested, a short information pertaining to each one has been added.

 Other minor issues:.

Line 180- define SMs if they are secondary metabolites, the extension for is used in line 181 so the abbreviation is not necessary

Response: Done

Fig. 1-You should cite the source if it is not an original graphic, and the state the program used to create the figure.

 Reponse: The figure is original and has been prepared by the authors. 

Authors should clarify the origin of secondary metabolites. For example, there are 4 sections on diosgenin, but it is not stated that diosgenin is present in every plants. If not, it should be included the specie. The same applies to solamargine and solanine etc

Response: They have been reported from almost all members of the solanaceae.

Line 528 “from 10 mg/ml to 0.0196 mg/ml” the smallest number should be the first. Or do you mean 10 µg/ml?

Response: Done

Table 2: Have these compounds been identified in all plants of the Solanaceae family? If not, the plant should be added to the table.

Reponse: They have been reported from almost all members of the Solanaceae family.

 In line 628-Ec50 the units are missing.

Response: Done

Line 697- 5 gm/ml ?  do you mean mg?

Response: Done

Round 2

Reviewer 1 Report

Comments and Suggestions for Authors

Authors replied to the reviewer' comments, therefore manuscript can be accepted.

Comments on the Quality of English Language

The English language has been corrected.